# Atherosclerotic-Derived Endothelial Cell Response Conducted by Titanium Oxide Nanotubes

**DOI:** 10.3390/ma16020794

**Published:** 2023-01-13

**Authors:** Ernesto Beltrán-Partida, Benjamín Valdez-Salas, Martha García-López Portillo, Claudia Gutierrez-Perez, Sandra Castillo-Uribe, Jorge Salvador-Carlos, José Alcocer-Cañez, Nelson Cheng

**Affiliations:** 1Laboratorio de Biología Molecular y Cáncer, Instituto de Ingeniería, Universidad Autónoma de Baja California, Blvd. Benito Juárez y Calle de la Normal s/n, Mexicali C.P. 21040, Baja California, Mexico; 2Coordinación Clínica de Cirugía, Hospital General de Zona No. 30, Instituto Mexicano del Seguro Social (IMSS), Av. Lerdo de Tejada s/n, Mexicali C.P. 21100, Baja California, Mexico; 3Magna International Pte Ltd., 10 H Enterprise Road, Singapore 629834, Singapore

**Keywords:** cell topography, cell-nanointeractions, tissue engineering, endothelial mechanosensing, nanobiotechnology

## Abstract

Atherosclerosis lesions are described as the formation of an occlusive wall-vessel plaque that can exacerbate infarctions, strokes, and even death. Furthermore, atherosclerosis damages the endothelium integrity, avoiding proper regeneration after stent implantation. Therefore, we investigate the beneficial effects of TiO_2_ nanotubes (NTs) in promoting the initial response of detrimental human atherosclerotic-derived endothelial cells (AThEC). We synthesized and characterized NTs on Ti6Al4V by anodization. We isolated AThEC and tested the adhesion long-lasting proliferation activity, and the modulation of focal adhesions conducted on the materials. Moreover, ultrastructural cell-surface contact at the nanoscale and membrane roughness were evaluated to explain the results. Our findings depicted improved filopodia and focal adhesions stimulated by the NTs. Similarly, the NTs harbored long-lasting proliferative metabolism after 5 days, explained by overcoming cell-contact interactions at the nanoscale. Furthermore, the senescent activity detected in the AThEC could be mitigated by the modified membrane roughness and cellular stretch orchestrated by the NTs. Importantly, the NTs stimulate the initial endothelial anchorage and metabolic recovery required to regenerate the endothelial monolayer. Despite the dysfunctional status of the AThEC, our study brings new evidence for the potential application of nano-configured biomaterials for innovation in stent technologies.

## 1. Introduction

Cardiovascular diseases (CVD) are the predominant pathophysiological processes causing mortality and morbidity, mainly in top-income countries [1,2,3]. Interestingly, among CVD, atherosclerosis points is the leading vascular condition characterized by critical chronic inflammation, lipids accumulation, fibrous elements, calcification, and plaque formation in the intima of blood vessels [4,5]. Moreover, atherosclerosis lesions alter and damage the endothelium activity close to the plaque boundaries. Far more critical is when a rupture of the atherosclerotic plaques takes place, exacerbating myocardial infarctions, strokes, and in the worst scenario, death. The catheter-based angioplasty followed by stenting implantation is the gold standard intervention for the revascularization and treatment of atherosclerotic lesions [6,7]. However, the angioplasty and the stent colocation process led to significant damage and alterations of the surrounding endothelial cells [8,9,10]. In addition, current studies have evidenced a cellular turnover accompanied by senescent transformed endothelial cells after implant colocation [7,11,12]. Consequently, the detrimental endothelial integrity conducts dysfunctional adhesion, proliferation, and monolayer cellular formation [13], prerequisites required to integrate the surface of the stent with the endothelial layer. Far more important is the subsequent complex clinical late thrombosis that can occur due to the lack of endothelialization or growing activity of new endothelial cells in the inner stent wall [14]. The low recovery of endothelial cells on the surface of the stent is considered the primary reason for late in-stent thrombosis because the nude exposed stent surface acts as a nucleation site for developing a thrombus [15].

The surface properties of the stent-designed materials have been demonstrated to play a pivotal role in the re-growth and subsequent re-endothelialization process after the initial hours of implant collocation [16,17,18,19]. Thus, different surface modification strategies have been applied in order to improve endothelial cell adhesion and proliferation. Interestingly, Hu et al. proposed that the vascular endothelial growth factor (VEGF) immobilization by using a heparin-bind strategy over Ti-based materials can improve the proliferation of healthy endothelial cells [20]. Even more, Bruni et al. suggested that thermal transformation and glow-discharge of Ti6Al4V increase the biocompatibility of human umbilical vein endothelial cells (HUVEC) under standard culture conditions [21]. Furthermore, Treves et al. investigated the effect of different heat treatments on Ti materials for vascular applications [22]. Similarly, an advanced electropolymerization coating method for incorporating dopamine into cardiovascular stents improved the proliferation of primary HUVEC of newborn healthy umbilical cords [23]. Despite those studies providing evidence of the innovation in surface strategy design to improve re-endothelialization, the vast majority of research uses healthy cell cultures instead of injured models. Therefore, atherosclerotic lesions require closely-related cellular conditions to ascertain the beneficial effect of surface modification technologies.

It has been reported that the nanostructured counterpart of vascular-based implants significantly improves the wound-healing behavior of vascular cells. For instance, TiO_2_ nanofibrous surfaces have been demonstrated to increase the adhesion, proliferation, and angiogenic behavior of HUVEC more than the non-modified counterpart [24]. Additionally, Choudhary et al. showed that nanostructured Ti increases the cellular growing density of commercial rat aortic endothelial cells [25]. Furthermore, a previous work using TiO_2_ nanotubes enhanced the cellular mobility of bovine aortic endothelial cells [26]. In addition, our group demonstrated that TiO_2_ nanotubes improve the angiogenic markers expression of bovine coronary artery endothelial cells [27]. Those works of surface modification highlight the benefits of nanoconfiguration in tailoring endothelial growth, increasing angiogenic functionality checkpoints, and forming the strictly demanded endothelial monolayer. However, it is critical to highlight that many surface modifications for coronary stents apply HUVEC or healthy models of endothelial cells instead of primary injured site-specific vascular cells. Far more important, the origin of the endothelial cells can arrange different morphology, fenestration of cell layer, proliferation, and angiogenic responses to biomaterials surfaces [28]. Considering the above-stated information and the urgent need for more realistic endothelial conditions, we test the re-endothelialization of NTs surfaces using endothelial cells derived from severe peripheral artery disease.

Here, we report the NTs angiogenic proliferative activity via addressing endothelial adhesion, formation of focal adhesion points, and cell-surface intimate contact bonding with the nanostructures. In addition, given the importance of cellular attachment and structural configuration, we describe the nanoscale cellular membrane roughness regulated by the nanostructured and control surfaces. The results bring new knowledge in the biology and understanding of endothelial cells from atherosclerotic disease dictated by nanostructured surfaces for application in stent technologies.

## 2. Materials and Methods

### 2.1. Synthesis of TiO_2_ Nanotubes

The NT surfaces were synthesized by following our previous method [29,30]. The medical alloy Ti6Al4V disks (ASTM F-136, Supra Alloys Inc., Camarillo, CA, USA) of 15 mm diameter and 5 mm thickness were mechanically polished according to the metallographic procedure ASTM E3-11 using SiC emery paper (100–2000 grit) and 0.5 µm alumina to accomplish a mirror finish. Then, the samples were cleaned in an ultrasonic bath (Branson, MO, USA) with acetone, ethanol, and distilled water for 30 min each. The Ti6Al4V samples were positioned on a flat 125 mL electrochemical cell and anodized using an electrolyte solution prepared with Microdacyn 60 (Oculus Technologies, Guadalajara, JAL, Mexico), 10 mg/mL NH_4_F (Sigma-Aldrich, St. Louis, MI, USA), and 100 mg/L NaCl (Sigma-Aldrich, St. Louis, MI, USA) at pH 6.8. Then, a 20 V potential was applied for only 5 min using a DC power supply and a platinum mesh as a counter electrode. The anodized materials were ultrasonically cleaned for 5 min in distilled water, rinsed with isopropyl alcohol, and dried in a desiccator for 12 h. The NTs diameter distributions were counted and measured using the ImageJ software (1.48v, NIH, Bethesda, ML, USA). Flat polished Ti6Al4V samples without any modification were used as control.

### 2.2. Surface Characterization

The surface morphology of the anodized Ti6Al4V samples was analyzed by Field Emission Scanning Electron Microscopy (FE-SEM; Tescan LYRA 3, Brno, Kohoutovice, Czech Republic) on random fields at 20 kV accelerating voltage. The Ti6Al4V flat materials were previously analyzed, thus confirming the smooth surface configuration [31]. The energy dispersive X-ray spectroscopy (EDX; Bruker, XFlash 6I30, Billerica, MA, USA) coupled to the FE-SEM at 10 kV with a large spot size to adjust a suitable count rate per second for spectrum collection. The surface topography of the NTs and Ti6Al4V was characterized using atomic force microscopy (AFM; NX10, Park Systems, Suwon, Republic of Korea). The AFM was equipped with an anti-acoustic box and an active vibration isolation table to prevent noises and vibrations that can affect the measurements. Moreover, the AFM was configured with a PPP-NCHR tip (Park Systems) with force constant = 42 N/m, resonance frequency = 330 kHz using the non-contact mode. The sample materials were fixed on a magnetic sampler holder using bi-adhesive carbon tape. The operation scan rate for all the samples was 0.5 Hz, with a Z-feedback set point of 3 × 10^3^ and amplitude of 21.376 × 10^3^ nm. The scan surface area was 2.5 µm^2^. To provide the surface roughness differences between the NTs and Ti6Al4V, we measured the average roughness (Ra), the square root mean (Rq) values, as well as the Z-profile graph, is provided [32]. The wettability of the experimental materials was assessed by the static water contact angle (WCA) by depositing a 5-µL droplet of deionized water at 20 ± 2 °C and 45% relative humidity. The droplet morphology was imaged using a high-performance CCD camera of an automatized tensiometer (Theta Attension, Biolin Scientific, Västra Frölunda, Sweden) equipped with an X-Y syringe. The WCA values were obtained using the ONE Attension software, which enables a highly precise analysis of the two angles of the drop. The surface crystalline configuration of the experimental materials was analyzed by means of X-ray diffraction (XRD) using a Bruker D8 Advanced diffractometer operated at 30 kV and 30 mA.

### 2.3. Isolation of Endothelial Cells from a Human Atherosclerotic Vessel

The present study included one patient (Table 1) with severe peripheral artery disease. Segments of the human femoral artery (2 to 5 cm) were dissected from patients accomplishing the inclusion criteria of being diagnosed with grade IV critical ischemia according to Fontaine classification [33]. The atherosclerotic segments of the superficial femoral artery were isolated by supracondylar amputation under regional anesthesia, followed by ligation and surgical excision. The segments were placed in 1 × phosphate-buffered saline (PBS) containing 1% penicillin/streptomycin (PS; Gibco-Invitrogen, Waltham, MA, USA) and immediately prepared for primary culture. The Ethical and Research Committee of the National Scientific Committee from Instituto Mexicano del Seguro Social approved the research protocol (Ref. R-2019-785-035; IMSS, Mexico City, Mexico). According to the Declaration of Helsinki, each participant signed informed consent to donate the explant for research purposes.

The primary culture of atherosclerotic endothelial cells (AThEC) was isolated from the whole luminal surface of the vessel. The vessel was washed several times with PBS in order to remove clots and blood and dissected into sections of 0.5 ± 0.1 × 0.2 ± 0.1 × 0.3 ± 0.1 mm. Then, the vessel fragments were placed in a sterile centrifuge tube containing 0.12% trypsin/0.02% EDTA in PBS and incubated at 37 °C for 90 min. The enzymatic reaction was stopped using medium 199 (M199; Gibco, Invitrogen, Waltham, MA, USA) supplemented with 10% fetal bovine serum (FBS; Gibco, Invitrogen). The solution was collected into a 15 mL centrifuge tube and washed with M199. The cellular suspension was then centrifuged at 2000 rpm for 10 min; the obtained pellet was resuspended in 5 mL of M199 with 10% FBS and 1% PS to be seeded into a 50 mm tissue culture plate. The cells were incubated at 37 °C, 5% CO_2_, and 95% air with medium changes every 48 h until reaching an 80% confluence. For the experimental analyses, the starting passages were from 2–4 to reduce the adverse effects of cellular senescence due to the origin of the cells.

### 2.4. Atherosclerotic Endothelial Cell Characterization

The AThEC were seeded at a density of 1 × 10^4^ cells/mL on glass slides pretreated with 0.1% gelatin solution (Sigma-Aldrich, St. Louis, MI, USA) and cultured for 24 h. The cells were washed thrice with PBS for 5 min and fixed with 4% paraformaldehyde (PA) for 30 min at room temperature (RT). Then, the cell substrates were permeabilized using 0.1% Triton X-100 in PBS for 30 min. The samples were washed thrice, incubated for 2 h at RT in bovine serum albumin (BSA) blocking solution (1% BSA/1 × PBS), and washed with PBS. The primary antibody for von Willebrand factor (1:500, Novusbio, Englewood, CO, USA) directly conjugated with DyLight 488 was incubated in blocking solution at 4 °C overnight and washed with PBS. The glass slides were mounted with coverslips containing fluorescence mounting medium (Fluroshield, Sigma-Aldrich, St. Louis, MI, USA), examined, and imaged using a green filter of an epifluorescence microscope (BX43, Olympus, Center Valley, PA, USA) equipped with a dark-field illumination system (Cytoviva^®^ 150 Resolution Imaging System, Auburn, AL, USA) (Appendix A).

### 2.5. Cell Culture

The primary AThEC were cultured in Dulbecco’s Modified Eagle’s Medium (DMEM, Gibco-Invitrogen) supplemented with 10% SBF and 1% PS. The experimental materials (NTs and Ti6Al4V control alloy) were placed in individual wells of a 12-well polystyrene tissue culture plate (Corning, Corning, NY, USA) to analyze the proliferation, morphology, and cell-ultrastructure [27]. The cells were seeded on the specimens at a cell density of 1 × 10^4^ cells/mL and incubated for different culture periods. 

### 2.6. Cell Viability by MTT

The metabolic activity of the primary atherosclerotic endothelial cells was studied using the 3-(4,5-dimethylthiazol-2-yl)-2,5-diphenyltetrazolium bromide (MTT) assay after 1, 3, and 5 days of culture. The cells were washed thrice with warm PBS, and 1 mL of MTT (Sigma-Aldrich, St. Louis, MI, USA) in DMEM (5 mg/mL) was added into each well and incubated at 37 °C in a humidified 5% CO_2_ incubator for 3 h. The formazan crystals resulting from the metabolic reaction were dissolved after discarding the medium containing MTT and transferring the 12-well plate into an orbital shaker at 200 rpm, 37 °C with dimethyl sulfoxide (Sigma-Aldrich, St. Louis, MI, USA) for 20 min. The dissolved crystals were then deposited into a 96-well polystyrene plate (Sigma-Aldrich, St. Louis, MI, USA), and the optical density (O.D.) was recorded at 590 nm using a microplate reader (Thermoskan, Thermo Fisher Scientific, Carlsbad, CA, USA).

### 2.7. Immunofluorescence Staining

The AThEC phenotype and the formation of focal adhesions were evaluated by means of F-actin filaments organization and convergence of Vinculin receptors conducted by the experimental surfaces [27]. The samples were washed three times with PBS and fixed in PA for 30 min at RT [32]. Next, the cells were washed, permeabilized using 0.1% TritonX-100 in PBS for 20 min and washed thrice. The samples were then incubated in blocking solution, washed thrice, and incubated with Alexa Fluor 488 phalloidin 1:100 dilution (Invitrogen, Carlsbad, CA, USA) in blocking solution for 1 h in order to analyze the F-actin fibers. For Vinculin localization, the samples were treated for 2 h with the primary antibody to Vinculin 1:100 dilution (Abcam, Cambridge, MA, USA) in the blocking solution at 4 °C. Then, the samples were washed, and Alexa Fluor 594 labeled anti-mouse 1:1000 dilution (Invitrogen, Carlsbad, CA, USA) was used as the secondary antibody for 1 h at RT. Next, the cell nuclei were counterstained using 4′,6′-diamidino-2-phenylindole (DAPI) (Molecular Probes, Carlsbad, CA, USA) in PBS, incubated for 20 min at RT, and washed three times with PBS. Finally, the materials were inverted and mounted onto coverslips with Fluroshield, analyzed, and photographed using a green (F-actin), red (Vinculin), and blue (DAPI) filter employing a fluorescence microscope (ZOE, Bio-Rad, Irvine, CA, USA) under similar magnifications. We captured 5–10 micrographs of each sample using the same exposure time to measure the fluorescence intensity of Vinculin. The average intensity was measured using ImageJ software from five random cells on each surface.

### 2.8. Atherosclerotic Endothelial Cells Characterization by FE-SEM

In order to analyze the morphological behavior and the cell-materials surface interactions conducted by the experimental materials, FE-SEM technology was applied as previously described [32]. The AEC were conditioned after 4 h, and 24 h of culturing by rinsing thrice with PBS (5 min) and fixed in 2.5% *w/v* glutaraldehyde (Sigma-Aldrich, St. Louis, MI, USA) buffered with 0.1 M sodium cacodylate (Sigma-Aldrich, St. Louis, MI, USA) a 4 °C overnight. The samples were then washed three times for 5 min in 0.1 M sodium cacodylate buffer and postfixed with 2.5% glutaraldehyde for 2 h at RT. Next, the cells were dehydrated in graded series of ethanol solutions (25%, 50%, 75%, and 100%) for 15 min at each concentration. Finally, the specimens were sputter-coated with gold (10-nm gold layer) for 8 s and characterized at 5 kV accelerating voltage. 

### 2.9. Endothelial Topography Analysis by AFM

For the analysis of the ultrastructural cell surface topography behavior of AThEC directed by the experimental materials at 4 h, we applied AFM. The AThEC were prepared following the FE-SEM protocol for morphology characterization without cellular dehydration and gold layer deposition process. The substrates were placed in an AFM equipped with the anti-acoustic box and characterized using a PPP-NCHR tip with force constant = 42 N/m, resonance frequency = 330 kHz using the non-contact mode. The operation scan rate for all the cellular topographies was 0.2 Hz, and the amplitude of 18.79 × 10^3^ nm. The scan surface area was 25 µm^2^. To provide the surface roughness differences between the NTs and Ti6Al4V cultured cells, we measured the average roughness (Ra) and the root mean square (Rq) values. The Z-profile graph is provided.

### 2.10. Statistical Analysis

Three independent experiments were performed, each in triplicate. The numerical data evaluation was conducted by one-way analysis of variance (ANOVA) followed by Tukey’s multiple comparison test when appropriate and two-tailed unpaired Student’s *t*-test [34]. A *p* < 0.05 was considered statistically significant.

## 3. Results and Discussion

The atherosclerotic lesions dictate extensive endothelial damage triggering a deficient re-endothelialization, cell adhesion, and proliferation to endovascular prosthetic surfaces, senescence, and apoptosis [1,7]. Far more concerning is the extensive research applying healthy endothelial models to test material surfaces (such as Ti6Al4V) without considering relevant conditions that recapitulate the damage of the site-specific cells. However, recent studies have suggested that nanostructured Ti6Al4V materials with NTs can improve the endothelial response compared to non-modified materials [26,27]. Similarly, our group described the beneficial relevant effects of NTs in promoting bone-growing functionality under detrimental diabetic conditions [32]. Therefore, considering the above-stated knowledge gaps, the present study reports the improved initial response of AThEC stimulated by Ti6Al4V anodized with NTs.

Figure 1 shows the surface characterization of the experimental materials, depicting the formation of self-organized, aligned, and homogenous NTs of 52 ± 6 nm in diameter (Figure 1a). Furthermore, the inset illustrates the honeycomb ordering of the NTs layer. Moreover, Figure 1b describes the XRD analysis without evidence of crystallographic patterns associated with anatase and rutile configurations. Thus, an amorphous surface layer was obtained [30]. The topography of the control Ti6Al4V presented a flat and smooth surface characterized by a regular Z-profile mapping closely linear to the baseline (Figure 1c). Similarly, the Ra (1.038 ± 0.11 nm) and Rq (1.43 ± 0.16 nm) values underline significantly more reduced roughness than the NTs, which resulted in 9.696 ± 0.41 nm of Ra and Rq 13.43 ± 1.26 nm, as expected. On the other hand, the nanoconfiguration exhibits irregular patterns of valleys and peaks with extensive reproducible tube-like arrangements, supporting the FE-SEM results (Figure 1a). The chemical analysis revealed an increased oxygen level after the anodization due to the growing thickness in the oxide layer of the NTs [35]. Importantly, electrochemical anodization generates a controlled passivation process, resulting in a significant increment of the protecting oxide layer of metallic materials, such as those of Ti-based alloys [36,37]. On the contrary, EDX detection limits were not able to sense the low oxygen content of the control (Figure 1d,e). The WCA reveals important information regarding the biocompatibility and cell adhesion behavior of a biomaterial surface. Hence, WCA displayed the fabrication of a nanostructured hydrophilic coating, which could be explained by the removal of organic pollutants during anodization and the inherent resulting oxygen levels [38].

In order to evaluate the metabolic activity of the AThEC over the experimental materials, we applied the MTT assay (Figure 2). Initially, it was evident that an early double-fold increase in the proliferation activity of AThEC after 24 h of culture, highlighting a significant change between the NTs and the control. It is essential to underline that the initial hours after stent colocation and direct implant contact with the neighboring cells are crucial in order to conduct cellular adhesion and proliferation [39]. Moreover, previous studies have advocated that the newly formed cell layer over the engraftment is mandatory to avoid side effects such as restenosis or early thrombosis [7]. Therefore, we followed up on the growth behavior of the AThEC for 3 and even 5 days. The results showed that the NTs sustain significantly higher cellular activity than the control alloy. However, we must focus on the decreased metabolic action between the NTs in the early and the lasting culture days. This interesting phenomenon can be explained by a faster initial endothelial proliferation rate followed by a subsequent decreased activity [27]. Cell cultures require enough surface area to proliferate rapidly on a material. Nonetheless, when the cellular monolayer reaches confluence, they enter a stationary phase, thereby downregulating the metabolic and proliferating activity [40,41]. On the other hand, the control alloy did not promote the growth of AThEC, suggesting that the damaged cells were not actively stimulated. This effect indicated that the atherosclerotic cells are exquisitely selective to the surface texturing conditions, as expected.

The cytoskeleton organization brings important information regarding the phenotype configuration stimulated by the physicochemical parameters of stent materials. Therefore, we characterized the F-actin distribution and formation of stress fibers after 24 h of incubation (Figure 3a). The results showed that the control alloy promoted an initial cellular extension, a width stretching among the surface, and reduced intracellular stress fibers accumulation. This effect could be explained due to the reduced surface area-to-volume ratio and decreased roughness of the flat alloy, as well as the enlarged cellular spreading, outlining complications for efficient cell migration [42]. However, the cellular morphology did not show the classical polygonal-like orientation appreciated for endothelial cells [43], despite the intercellular connections detected on the control alloy. Meanwhile, the NTs conducted an irrefutable formation of stress fibers accompanied by extensive interconnections, aligned endothelial orientation, and a rhomboid-like configuration, significantly contrasting to the control surface. Far more important is the hallmarking endothelial overexpression of vinculin harbored by the NTs (Figure 3b). Interestingly, the activation of vinculin, a receptor-dependent protein participating in forming focal adhesions, was far more disseminated in the membrane boundaries of the AThEC growing on the NTs. Furthermore, the intercellular convergence between vinculin and F-actin fibers was evidently promoted by the NTs, thus highlighting the formation of focal adhesions. The findings may be elucidated by the supported cytoplasmic crisscross pattern of stress fibers organization which are involved in cellular mobility, proliferation, and mitochondrial activity [44]. The retroactive vinculin converging within those filaments develops consistent focal adhesions strongly associated with the phosphorylation of the focal adhesion kinase (FAKs) and the subsequent signaling pathway [45]. Interestingly, our current results align with a previous work of damaged endothelial cells isolated from human diabetic artery walls cultured over micro- and nano-patterned polydimethylsiloxane surfaces [46]. The authors discovered that the metabolic status of the endothelial cells mainly regulated the cytoskeletal arrangement conducted over different surface topographies, thus suggesting that healthy endothelial cells behave in a similar topographical manner. Meanwhile, the diabetic cells decreased the formation of stress fibers and focal adhesions under anisotropic topographies. In this regard, FAKs are intimately involved in robust cell adherence to materials surfaces, cell survival, and long-lasting endothelial propagation [45,47], information that supports our outcomes of metabolic activity (Figure 2). The current trends following our results are in accordance with the FE-SEM shreds of evidence of the initial AThEC adhesion at 4 h, as depicted in Figure 4. Thus far, the NTs worked as guiding platforms for the AThEC anchorage and formation of well-defined cellular bodies (purple square) with long cellular connections, and the growth of thick, protuberant fibrillary-like filopodia (white triangles). Nonetheless, the Ti6Al4V control orchestrated a flat and smooth cell layer adhesion showing the extension and displacement of thin translucent filopodia. Therefore, the control material could not influence the progressive cell adhesion required for the initial stent re-endothelialization, as counter-evidenced by the NTs.

Figure 5 shows that after 24 h, the flat control material deprived the endothelial extension characterized by a poor-spread morphology, deficient cell–cell interconnections, and a flat translucent cell-body structure. Moreover, we detected smooth and disrupted cell filopodia, suggesting that the diseased cells cannot sustain a continued cellular bonding to the control. On the other hand, the NTs displayed outstanding development of a higher number of thicker filopodia microspikes anchoring to the surface and performing cell–cell continuous web interconnections. Furthermore, the AThEC showed an evident growth of cellular bodies with underlining edges, spreading over the nano-surface conducting to the formation of a monolayer, which is a mandatory prerequisite for the regeneration of new and functional endothelial cells [48]. Far more interesting are the high-zoom micrographs that evidenced widespread lamellipodia guided by the NTs (Figure 5). Given these significant results, we performed high magnifications in order to elucidate the cell–nanostructure surface interactions modulated at the nanoscale level (Figure 6). Interestingly, the NTs stimulated a striking secretion and deposition of extracellular matrix (ECM) at the boundaries of the AThEC and the NTs (white arrow in Figure 6b). The ECM extensively coated the NTs, which was more evident at higher amplifications. This interesting condition can be explained by a triggered cellular adaptation carried out by the NTs after establishing good cell adhesion and an adequate microenvironment [32]. Likewise, we can identify that the AThEC developed strikingly thick binding filopodia that are directly interconnecting with ECM coating and the NTs at the nanoscale level (Figure 6d,e). In this regard, in the inset is evidenced a fusion between the filopodia-ECM-NTs system (yellow arrows in Figure 6d). Our findings suggest that the damaged and senescent AThECs are integrating and forming a synthetic “vascular coat” that could be indicative of endothelial functionality. These interactions can be orchestrated thanks to the much surface area-to-volume ratio provided by the NTs. Therefore, we hypothesize that the extensive interplay between the pores structures, the stable deposition of ECM-associated proteins [26], the formation of focal adhesions, and the intra- and extracellular transport of nutrients and signaling molecules [49] facilitated by the NTs, can explain the improved AThEC response. However, more experiments are required to measure those differences.

Endothelial cell migration is critical during angiogenesis, vascular remodeling, bypass procedures, and, more importantly, for re-endothelialization after angioplasty and stent colocation [50]. Moreover, endothelial migration includes protrusions of the leading edges, contractility of the cells, reorganization of the cytoskeleton, and establishment of focal adhesions [51], as observed here. Furthermore, several studies described that the cell surface roughness and height, the topographical extension of the adhered and migrating cells, control the mechanomodulation collectively with the surfaces of the materials [52,53]. Therefore, we performed an ultrastructural analysis of the damaged AThEC growing on the experimental materials, employing AFM in order to elucidate the nanostructural cell topography (Figure 7). Interestingly, Figure 7a illustrates the 2D and 3D cell orientation showing the formation of an extended cell body and an irregular membrane topography conducted by the NTs. On the other hand, the Ti6Al4V depicted the growth of a thinner and flat cellular body, illustrating more delicate and shorter filopodia, as previously detected (Figure 5). To analyze the cell surface roughness stimulated by the materials, we measure the Ra and Rq values corresponding to the cell body topography. The NTs significantly reduced the arithmetic and mean square root roughness cellular membrane compared with the Ti6Al4V (Figure 7b,c). Furthermore, the roughness differences were accompanied by extensive variations in the cell height and the valley-to-peak curves. Further highlighting constant and more regular topography extended by the nanostructured coating (Figure 7d). These results suggest the formation of a widespread monolayer. Our current results are in line with a recent work proposing that increased cellular senescent can be proportional to the membrane roughness and cellular height [54]. The authors suggested that microvascular endothelial cells exposed to the neurotoxic amyloid β (Aβ1-42) oligomers induce a senescence phenotype accompanied by altered cell membrane roughness and height [54]. Similarly, it was concluded that the nanostructured surfaces improved membrane roughness and height orientation can guide higher cellular locomotions and recovery from senescent cellular activity [55]. Interestingly, the stimulation of focal adhesion has been closely related to a higher adhesion force, nanomechanical modulation, improved roughness, and stiffness behavior [56], as evidenced by our AFM and fluorescence analysis (see Figure 3 and Figure 7). Nonetheless, more work on cellular nanomechanics is required to elucidate the underlying mechanisms.

The formation of atherosclerotic plaque severely alters the biochemical and physiological function of the endothelial cells [4,7]. In our present work, we isolated damaged AThEC from the most severe peripheral artery disease, which showed outcomes of cellular senescence, reduced proliferative activity, and a limited number of cell passages. Therefore, it was difficult to obtain a viable culture that allowed the preliminary evaluation presented herein. Previous works have suggested that the endothelial cells from direct atherosclerotic lesions can derive into senescent and damaged endothelial cultures [57,58]. For instance, Cho et al. suggested that the endothelial layer of the atherosclerotic sections of human carotids upregulated the CD9 tetraspanin protein, which is intimately involved in senescent control and plaque development [59]. Moreover, it was reported in an in vitro study that vascular endothelial cells from atherosclerotic lesions exhibited high levels of senescence-associated β-gal activity and telomere shortening [60]. These studies of damaged cellular response are in line with the detrimental metabolic activity and deficient proliferation detected in our isolated AThEC. Similarly, it was reported that endothelial cells isolated from human diabetic arteries are generally less responsive and exhibited decreased endothelial functionality, rather than the healthy cells cultured under identical surface and microenvironment conditions [46]. Despite the deteriorated status of the primary AThEC, we can highlight that the NTs tailored focal adhesions, intimate surface bond-contact interactions, and beneficial mechanosensing performance. However, we recommend further studies regarding angiogenic activity, AThEC derived from more patients and different NTs diameters in order to extend the contributions of these findings.

## 4. Conclusions

Here, we present the initial response of damaged and senescent human-derived AThEC conducted by NTs and the non-modified surface. Our work highlights that the nanostructured surfaces improve the initial adhesion, the formation of cytoskeleton stress fibers, and improved focal adhesion in the damaged cells. Moreover, the results of MTT indicated that the NTs could promote a faster proliferation of the AThEC and a long-lasting metabolic activity after 5 days of being cultivated. Furthermore, the NTs guided intimal cell-surface contact interactions tailoring an extensive ECM deposition as observed by the high-resolution FE-SEM analyses. These beneficial results are partly explained by the promoted focal adhesions and guided filopodia anchorage by the NTs. Importantly, the subsequent cell-surface roughness and the cellular stretching behavior harbored by the nanoconfigured materials can explain part of the mechanosensing stimulation harbored by the NTs. Despite the senescent and dysfunctional status of the AThEC, the NTs could mitigate those adverse conditions for potential applications for stent surface technologies.

## Figures and Tables

**Figure 1 materials-16-00794-f001:**
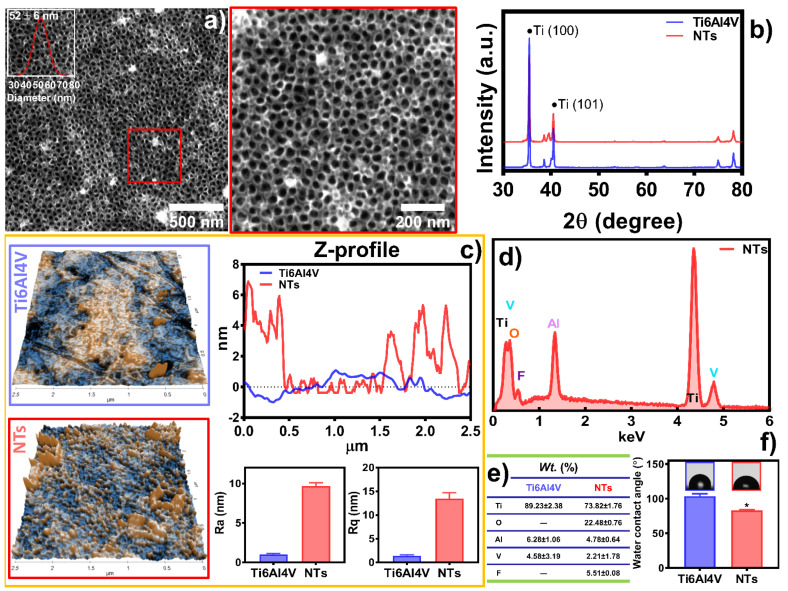
Physicochemical characterization of the experimental surfaces: (**a**) FE-SEM of the NTs, the red square highlights a high-magnification area. The inset shows the diameter distribution of the NTs; (**b**) XRD analysis of the substrates; (**c**) Topography evaluation by AFM illustrating the materials’ Ra, Rq, and Z-profile graph; (**d**) Elemental analysis EDX spectra; and (**e**) chemical quantification of the substrates; (**f**) WCA of the surfaces. The * shows significant differences.

**Figure 2 materials-16-00794-f002:**
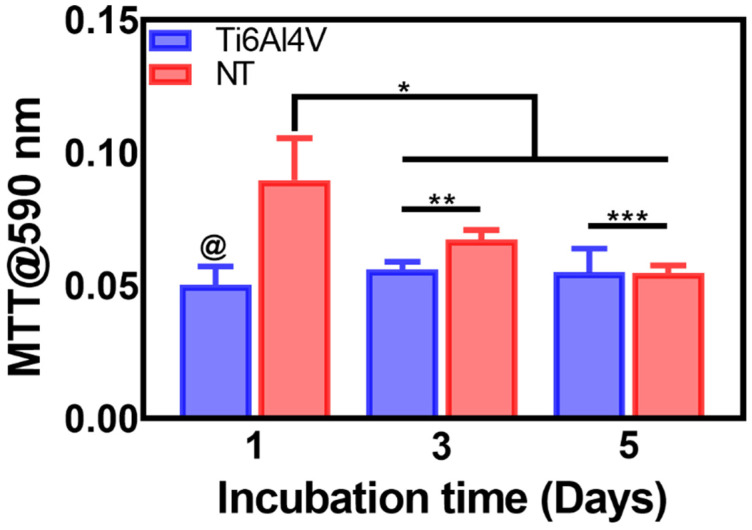
MTT assay of the AThEC proliferation on the Ti6Al4V and NTs surfaces after 1, 3, and 5 days of culture. The @ represents significant differences between the materials after 1 day of incubation. The * shows significant changes among NTs of 1 day and the proliferation analyses at 3 and 5 days of growth. The ** and *** indicate increased proliferation rate over the NTs surfaces.

**Figure 3 materials-16-00794-f003:**
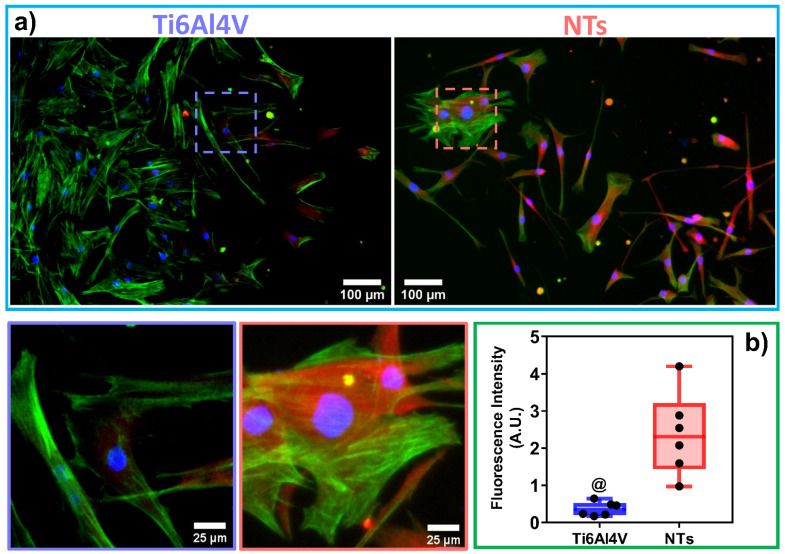
Immunofluorescence of vinculin (red) and F-actin patterning (green) stimulated by the surfaces at 24 h of culture: (**a**) AThEC showing the vinculin expression among the cells, the endothelial phenotypic orientation, and the formation of cytoskeleton stress fibers conducted by the materials. The dotted squares illustrate a high zoom highlighting the formation of focal adhesion zones; (**b**) Graph of vinculin fluorescence expression, dots represents the individual values. The @ indicates significant differences.

**Figure 4 materials-16-00794-f004:**
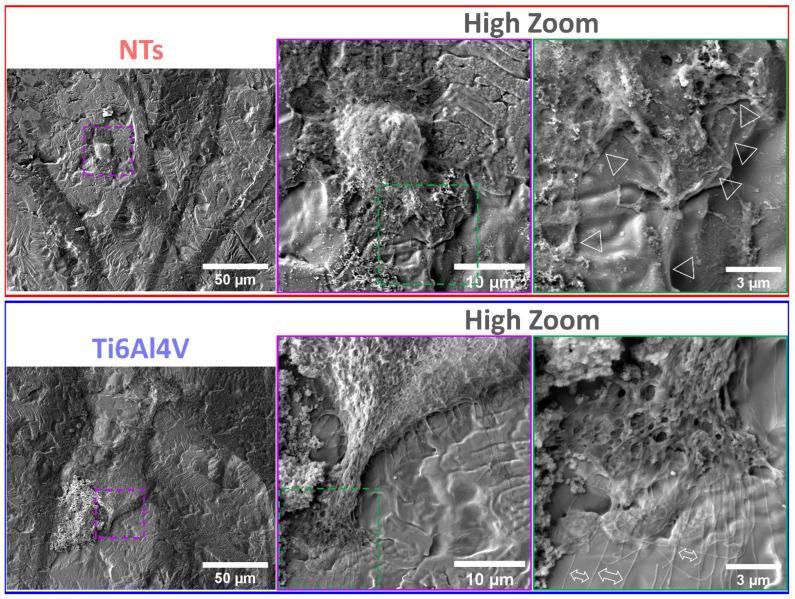
FE-SEM characterization of the AThEC morphology and behavior after 4 h of culture. The purple and green dotted squares represent the high zoom of the cell structures. The white triangles illustrate the contact adhesion zones of the AThEC on the NTs. The white double-headed arrows show the formation of thin filopodia.

**Figure 5 materials-16-00794-f005:**
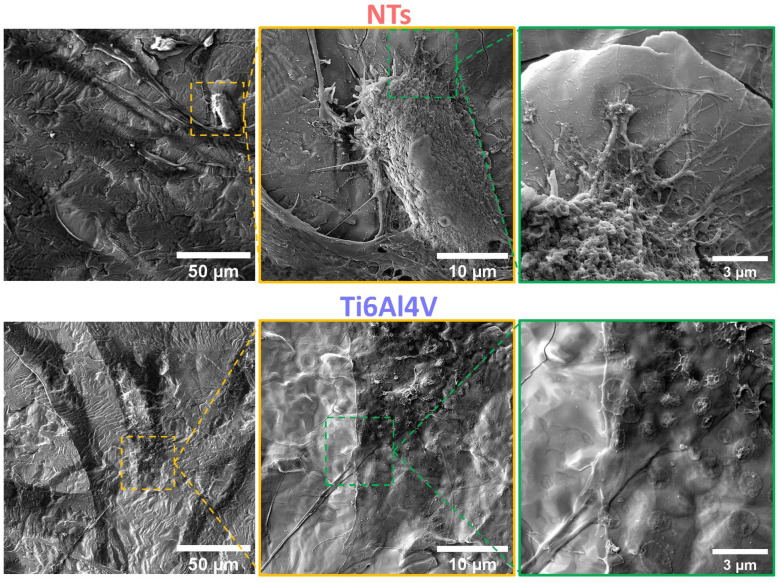
FE-SEM evaluation of the AThEC after 24 h of culture. The yellow dotted squares show the high zoom of the cell bodies. The green dotted squares illustrate the filopodia and contact adhesion zones of the AThEC to the surfaces.

**Figure 6 materials-16-00794-f006:**
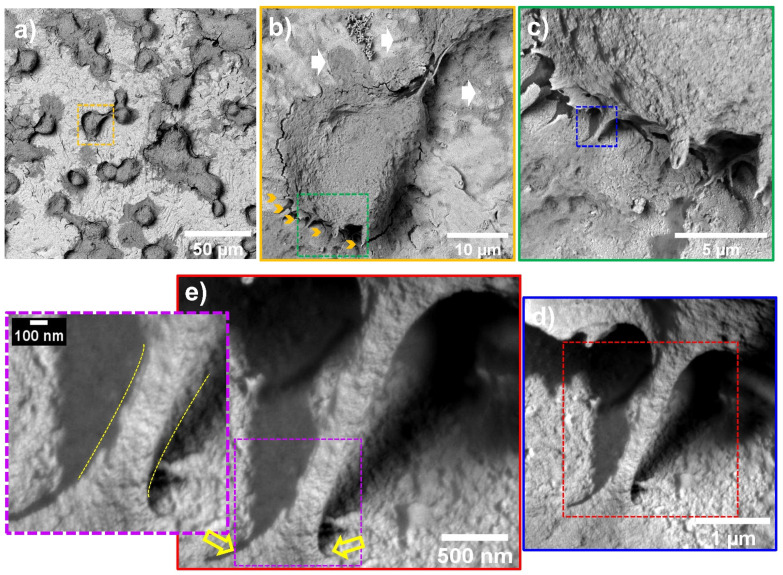
AThEC morphology and nanoscale cell-NTs contact interactions after 24 h of culture: (**a**) AThEC polygonal morphology conducted by the nanostructured surface; (**b**) Well-defined cell body formation illustrating the development of anchoring filopodia (yellow arrows) and ECM secretion (white arrows); (**c**) High zoom promoting the ECM and filopodia contacts with the NTs; (**d**) Filopodia’s intimate contact with the ECM and the NTs; (**e**) Nanoscale contact interface highlighting the filopodia intimate insertion with the NTs (yellow arrows). The purple-dotted squares show the filopodia elongation and thickness modulated at the nanoscale interface. The different dotted squares represents the magnification sequence. The yellow dotted line highlights the filopodia-surface nanoscale interface.

**Figure 7 materials-16-00794-f007:**
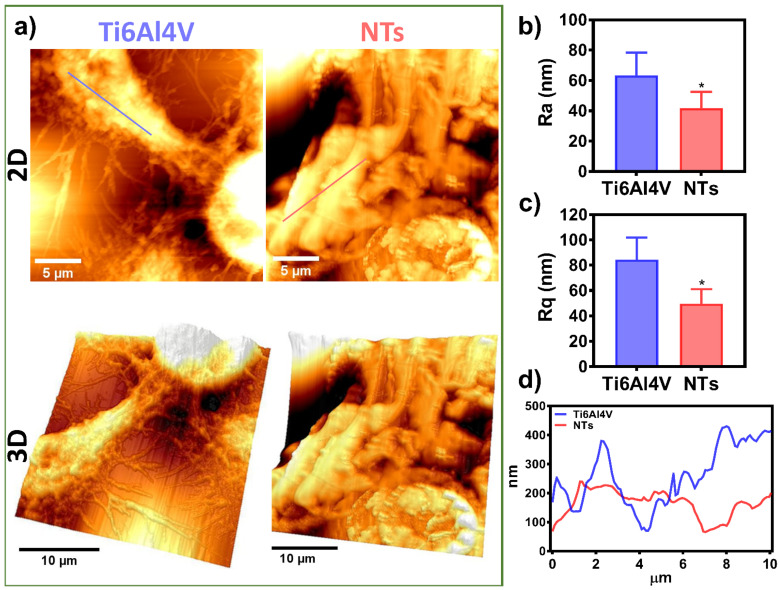
Cell topography analysis of the AThEC cultured on the experimental materials after 24 h: (**a**) 2D and 3D AFM micrographs of the ECS on Ti6Al4V and NTs; (**b**) Ra of the AEC; (**c**) Rq of the AEC; (**d**) Graph of the Z-profile showing the topographic map of the cells conducted by the surfaces. The * shows significant differences.

**Table 1 materials-16-00794-t001:** Clinical profile of the patient undergoing supracondylar amputation.

Age, Years	62
Gender	Male
Nicotine use (Smoker)	-
Comorbidity	
Type 2 Diabetes mellitus	10 years of follow up
Hypoglycemic drugs	Metformin
Hypertension	-
Dyslipidemia	-
Cerebrovascular disease	3 years of follow up
Blood analysis	
Glucose, mg/dL	89.1
Urea, mg/dL	56.3
Creatinine, mg/dL	1.7
Hemoglobin, g/dL	8.8

## Data Availability

Not applicable.

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
