# Peer review of "Atherosclerotic-Derived Endothelial Cell Response Conducted by Titanium Oxide Nanotubes"

_materials, 2023, doi:10.3390/ma16020794_

Round 1
Reviewer 1 Report
Reviewer’s Comments:
The manuscript “Atherosclerotic-derived endothelial cell response conducted by titanium oxide nanotubes” is very interesting work. This paper investigates the atherosclerosis lesions are described by the formation of an occlusive wall-vessel plaque that can exacerbate infarctions, strokes, and even death. Furthermore, atherosclerosis damages the endothelium integrity, avoiding proper regeneration after stent implantation. Therefore, we investigate the beneficial effects of TiO2 nanotubes (NTs) in promoting the initial response of detrimental human atherosclerotic-derived endothelial cells (AThEC). We synthesized and characterized NTs on Ti6Al4V by anodization. We isolated AThEC and tested the adhesion and long-lasting proliferation activity promoted by the surfaces. Furthermore, the cytoskeleton arrangement and the modulation of focal adhesions were studied on the materials. Moreover, ultrastructural cell-surface contact at the nanoscale and membrane roughness were evaluated to explain the results. However, the following issues should be carefully treated before publication.
1. In abstract, the author should add more scientific findings.
2. Keywords: the synthesized system is missing in the keywords. So, modify the keywords.
3. In the introduction part, the introduction part is not well organized and cited references should cite recently published articles such as 10.3389/fchem.2022.1023316, 10.3390/molecules27196457
4. Introduction part is not impressive and systematic. In the introduction part, the authors should elaborate the scientific issues in the endothelial cell response research.
5. Immunofluorescence Staining…, The author should provide reason about this statement “The samples were washed three times with PBS and fixed in PA for 30 min at RT”.
6. The authors should explain regarding the recent literature why “The substrates were placed in an AFM equipped with the anti-acoustic box and characterized using a PPP-NCHR tip with force constant = 42 N/m, resonance frequency = 330 kHz using the non-contact mode”.
7. Results and Discussion. The author should explain the latest literature “The chemical analysis revealed an increased oxygen level after the anodization due to the growing thickness in the oxide layer of the NTs”.
8. The author should provide reason about this statement, “The results showed that the control alloy promoted an initial cellular extension, a width stretching among the surface, and reduced intracellular stress fibers accumulation”.
9. Comparison of the present results with other similar findings in the literature should be discussed in more detail. This is necessary in order to place this work together with other work in the field and to give more credibility to the present results.
10. The conclusion part is very week. Improve by adding the results of your studies.
Author Response
Reviewer 1
Dear reviewer, thank you very much for your important recommendations and the experience shared in your comments; we much appreciate it. As per your comments, we have addressed the point itemized below. We believe that the present modifications would fulfill your essential indications. We hope you find our article as interesting as we do.
The manuscript “Atherosclerotic-derived endothelial cell response conducted by titanium oxide nanotubes” is very interesting work. This paper investigates the atherosclerosis lesions are described by the formation of an occlusive wall-vessel plaque that can exacerbate infarctions, strokes, and even death. Furthermore, atherosclerosis damages the endothelium integrity, avoiding proper regeneration after stent implantation. Therefore, we investigate the beneficial effects of TiO2 nanotubes (NTs) in promoting the initial response of detrimental human atherosclerotic-derived endothelial cells (AThEC). We synthesized and characterized NTs on Ti6Al4V by anodization. We isolated AThEC and tested the adhesion and long-lasting proliferation activity promoted by the surfaces. Furthermore, the cytoskeleton arrangement and the modulation of focal adhesions were studied on the materials. Moreover, ultrastructural cell-surface contact at the nanoscale and membrane roughness were evaluated to explain the results. However, the following issues should be carefully treated before publication.
- In abstract, the author should add more scientific findings.
Thank you for the recommendation, we have updated the abstract.
- Keywords: the synthesized system is missing in the keywords. So, modify the keywords.
We modified the keywords, thank you for the recommendation.
- In the introduction parte, the introduction part is not well organized and cited references should cite recently published articles such as 10.3389/fchem.2022.1023316, 10.3390/molecules27196457
We appreciate the recommendation, the new cites have been incorporated.
- Introduction part is not impressive and systematic. In the introduction part, the authors should elaborate the scientific issues in the endothelial cell response research.
Interesting recommendation the modifications have been included in the introduction section.
- Immunofluorescence Staining…, The author should provide reason about this statement “The samples were washed three times with PBS and fixed in PA for 30 min at RT”.
This is part of the procedure that is used in order to prepare the samples for immunofluorescence staining. First, it is important to wash the samples to remove the medium and residues of the cellular growing activity. Then, the cells need to be fixed using a chemical cross-linker such as paraformaldehyde to reduce the cellular alteration occasioned by the subsequent immunoreactions [1]. We include a reference supporting the procedure next to the statement in the new version of the manuscript.
- The authors should explain regarding the recent literature why “The substrates were placed in an AFM equipped with anti-acoustic box and characterized using a PPP-NCHR tip with force constant = 42 N/m, resonance frequency = 330 kHz using the non-contact mode”.
Thank you for the observation. Those are the optimal parameters for obtaining the AFM micrographs presented in the work. These parameters are the most optimal data found during the evaluations using the specifications and accessories equipped in our AFM.
- Results and Discussion. The author should explain the latest literature “The chemical analysis revealed an increased oxygen level after the anodization due to the growing thickness in the oxide layer of the NTs”.
We have included a brief statement explaining this observation.
- The author should provide reason about this statement, “The results showed that the control alloy promoted an initial cellular extension, a width stretching among the surface, and reduced intracellular stress fibers accumulation”.
We appreciate the recommendation; we included a brief statement explaining this observation.
- Comparison of the present results with other similar findings in the literature should be discussed in more detail. This is necessary in order to place this work together with other woks in the field and to give more credibility to the present results.
We appreciate this important consideration. We did our best to include closely related works associated with our present study. To the limit of our knowledge, few works evaluate the potential in vitro action of isolated human AThEC on nanostructured materials.
- The conclusion part is very week. Improve by adding the results of your studies.
Thank you for the suggestion. We modified our conclusion as per your recommendation.
References
- DiDonato, D.; Brasaemle, D. L., Fixation Methods for the Study of Lipid Droplets by Immunofluorescence Microscopy. Journal of Histochemistry & Cytochemistry 2003, 51, (6), 773-780.

Reviewer 2 Report
The authors demonstrated that TiO2 nanotubes (NTs) increase cell proliferation of endothelial cells extracted from a patient with peripheral artery disease (AThEC) and also enhance levels of vinculin. NTs also improves cell adhesion and surface properties of AThEC.
The work is interesting and suggests that NTs may represent a useful strategy for improve endothelial behavior in the presence of a stent implantation. However, several aspects of the manuscript should be improved
1) The characterization of the effects of NTs in AThEC isolated from only one patient is not sufficient since the genetic background of the patient may also influence the endothelial response to NTs. I suggest performing the experiments in at least another two patients, whenever possible. Otherwise, the authors should test the effects of NTs in commercially available human endothelial cell line undergoing atherosclerotic stress.
2) Fig. 4 and Fig. 5 are the same representative images, but the analyses were performed at different times (4 hrs and 24 hrs). Please check carefully and clarify this aspect.
3) SEM images are convincing. However, the authors should analyze at molecular levels changes in markers of cell adhesion, as well as of cell senescence.
4) Endothelial dysfunction during atherosclerosis is also characterized by the increase of oxidative stress and by the decrease of nitric oxide bioavailability. What are the effects of NTs on these parameters?
Author Response
Reviewer 2
Dear reviewer, we appreciate your constructive observations very much; thank you. As per your comments, we have addressed the point itemized below. We believe that the present modifications would fulfill your essential indications. We hope you find our article as interesting as we do.
The authors demonstrated that TiO2 nanotubes (NTs) increase cell proliferation of endothelial cells extracted from a patient with peripheral artery disease (AThEC) and also enhance levels of vinculin. NTs also improved cell adhesion and surface properties of AThEC.
The work is interesting and suggests that NTs may represent a useful strategy for improve endothelial behavior in the presence of a stent implantation. However, several aspects of the manuscript should be improved
1) The characterization of the effects of NTs in AThEC isolated from only one patient is not sufficient since the genetic background of the patient may also influence the endothelial response to NTs. I suggest performing the experiments in at least another two patients, whenever possible. Otherwise, the authors should test the effects of NTs in commercially available human endothelial cell line undergoing atherosclerotic stress.
Dear reviewer, this is an important observation that we agree with you. However, we performed several surgical procedures in order to obtain damaged blood vessels from patients undergoing amputation. We repeated the isolation procedures several times to obtain a sufficient number of cells to accomplish the objective of the present work. Unfortunately, as postulated in our manuscript (results and discussion section), it is very difficult to acquire a viable and pure culture of AThECs from each severely damaged vessel. Importantly, we agree that more studies, including many patients, are recommended to increase the statistical power and evaluate the genetic variability. On the other hand, including cellular conditions of atherosclerotic stress are interesting. Nonetheless, the application of commercial cell lines has the disadvantage of reducing the realistic proliferative and angiogenic behavior, avoiding the donor-to-donor variations required for population comparison and the inaccurate reflection of the site-specific endothelial reactions [1-3]. Furthermore, our work brings mounting evidence of the advantages provided by nanostructured materials (e.g., NTs) against a non-modified counterpart. This information and the cumulative results point toward the versatility of NTs for promoting the proliferation of damaged derived cells. Therefore, we communicate to you for considering our work as a preliminary study that brings new information on the applicability and importance of using endothelial cells of site-specific damaged zones for the characterization of the potential endothelial response of nanostructured materials for stent applications.
2) Fig. 4 and Fig. 5 are the same representative images, but the analyses were performed at different times (4 hrs and 24 hrs). Please check carefully and clarify this aspect.
Thank you for this critical observation; we apologize for this mistake. Please, find attached the corresponding Figure. The figure label is correct and corresponds to Figure 5.
3) SEM images are convincing. However, the authors should analyze at molecular levels changes in markers of cell adhesion, as well as of cell senescence.
This is an interesting observation that improves our work. In our previous work on endothelial functionality by synthesized NTs, we showed that growing endothelial cells of a healthy model improved the expression of cell surface receptors. Interestingly, the increased expression of vascular endothelial growth factor receptor 2 (VEGFR2) and even endothelial nitric oxide synthase was significantly promoted during the growing conduction on NTs [4]. Moreover, we described enhanced cell adhesion and the significant expression of vinculin, generating focal adhesions, which were in line with the elevated action of endothelial marker receptors. Furthermore, Pan et al. suggested that 30 and 50 nm NTs also promote the secretion of VEGF, thus far, resulting in better cellular adhesion, spreading, and proliferation [5]. Similarly, Tan et al. demonstrated that nanostructured TiO2 surfaces increase the expression of the cell surface markers VEGF, Von Willebrand Factor, and PECAM-1 of healthy HUEVEC cells [6]. Far more important, the authors communicated that HUVEC growing on the nanomodified surfaces extensively improved cellular adhesion, spreading followed by monolayer formation and a higher proliferation rate, as observed in our current work. Additionally, Cao et al. communicated that NTs surfaces, in comparison with the flat counterparts, significantly decrease the expression of VECAM-1, an adhesion surface receptor linked to vessel inflammation [7]. Interestingly, this study concluded that the NTs could reduce expression markers of vessel inflammation while promoting the phosphorylation of focal adhesion kinase. Thus, indicating that the promoted endothelial adhesion and monolayer formation are linked to high focal adhesion. Furthermore, considering the evidence of endothelial marker expressions, the increase of vinculin expression with higher endothelial adhesion improves the activation of endothelial surface markers. However, our current results bring important information to continue the study of the molecular expression of damaged atherosclerotic endothelial cells and senescence status guided by nanostructured materials. Nonetheless, due to the difficulty of obtaining and proliferating the AThEC, we are currently working on extending endothelial cells and conducting more molecular expression studies.
4) Endothelial dysfunction during atherosclerosis is also characterized by the increase of oxidative stress and by the decrease of nitric oxide bioavailability. What are the effects of NTs on these parameters?
We are in accordance with your observations. Atherosclerosis is characterized by a severe dysfunction of the endothelial cells regulating the vasculature and maintaining the dynamic balance of blood vessels. However, a previous work of anodized 50 nm NTs proposed that the nanostructured materials increased NO secretion after 1 and 3 days of continuous incubation [5]. Moreover, Brammer et al. conducted a front-line comparative experiment between NTs and flat control surfaces, suggesting that the nanoconfiguration orchestrated an amplified releasing of NO after 24h [8]. On the other hand, we described the increased expression of p-eNOS conducted by NTs in healthy endothelial cells [4]. Our findings suggested that the NTs increased surface area-to-volume ratio promoted the formation of focal adhesion (FA). Therefore, the FA up-regulates the mitogen-activated protein kinase (MAPKs) signaling pathway proposing that similar behavior is modulated in the AThEC response to NTs. Nonetheless, the present work aims to evaluate the initial behavior of human AThECs derived from critically damaged peripheral arteries on nanostructured surfaces in order to bring evidence of the importance of applying realistic cell lines for future studies.
References
- Hauser, S., F. Jung, and J. Pietzsch, Human Endothelial Cell Models in Biomaterial Research. Trends in Biotechnology, 2017. 35(3): p. 265-277.
- James Kirkpatrick, C., et al., Cell culture models of higher complexity in tissue engineering and regenerative medicine. Biomaterials, 2007. 28(34): p. 5193-5198.
- Chen, J., et al., Recent Progress in in vitro Models for Atherosclerosis Studies. Frontiers in Cardiovascular Medicine, 2022. 8.
- Beltrán-Partida, E., et al., Improved in vitro angiogenic behavior on anodized titanium dioxide nanotubes. Journal of Nanobiotechnology, 2017. 15(1): p. 10.
- Pan, C., et al., Improved Blood Compatibility and Endothelialization of Titanium Oxide Nanotube Arrays on Titanium Surface by Zinc Doping. ACS Biomaterials Science & Engineering, 2020. 6(4): p. 2072-2083.
- Tan, A.W., et al., Enhanced in vitro angiogenic behaviour of human umbilical vein endothelial cells on thermally oxidized TiO2 nanofibrous surfaces. Scientific Reports, 2016. 6(1): p. 21828.
- Cao, Y. and T.A. Desai, TiO2-Based Nanotopographical Cues Attenuate the Restenotic Phenotype in Primary Human Vascular Endothelial and Smooth Muscle Cells. ACS Biomaterials Science & Engineering, 2020. 6(2): p. 923-932.
- Brammer, K.S., et al., Enhanced Cellular Mobility Guided by TiO2 Nanotube Surfaces. Nano Letters, 2008. 8(3): p. 786-793.

Round 2
Reviewer 2 Report
The authors addressed my comments without performing additional experiments. However, I am satisfied.